# Breastfeeding Education and Support to Improve Early Initiation and Exclusive Breastfeeding Practices and Infant Growth: A Cluster Randomized Controlled Trial from a Rural Ethiopian Setting

**DOI:** 10.3390/nu13041204

**Published:** 2021-04-06

**Authors:** Misra Abdulahi, Atle Fretheim, Alemayehu Argaw, Jeanette H. Magnus

**Affiliations:** 1Department of Population and Family Health, Jimma University, Jimma 378, Ethiopia; yemarimwork2@gmail.com; 2Department of Community Medicine and Global Health, University of Oslo, 0316 Oslo, Norway; 3Faculty of Health Sciences, Oslo Metropolitan University, 0130 Oslo, Norway; atle.fretheim@medisin.uio.no; 4Centre for Informed Health Choices, Norwegian Institute of Public Health, 0473 Oslo, Norway; 5Department of Food Technology, Safety and Health, Faculty of Bioscience Engineering, Ghent University, B-9000 Ghent, Belgium; 6Faculty of Medicine, University of Oslo, 0316 Oslo, Norway; j.h.magnus@medisin.uio.no; 7Department of Global Community Health and Behavioral Sciences, Tulane School of Public Health and Tropical Medicine, New Orleans, LA 70112, USA

**Keywords:** community-based, peer support, breastfeeding initiation, exclusive breastfeeding, infant growth, breastfeeding knowledge, attitude

## Abstract

Although peer-led education and support may improve breastfeeding practices, there is a paucity of evidence on the effectiveness of such interventions in the Ethiopian context. We designed a cluster-randomized trial to evaluate the efficacy of a breastfeeding education and support intervention (BFESI) on infant growth, early initiation (EI), and exclusive breastfeeding (EBF) practices. We randomly assigned 36 clusters into either an intervention group (*n* = 249) receiving BFESI by trained Women’s Development Army (WDA) leaders or a control group (*n* = 219) receiving routine care. The intervention was provided from the third trimester of pregnancy until five months postpartum. Primary study outcomes were EI, EBF, and infant growth; secondary outcomes included maternal breastfeeding knowledge and attitude, and child morbidity. The intervention effect was analysed using linear regression models for the continuous outcomes, and linear probability or logistic regression models for the categorical outcomes. Compared to the control, BFESI significantly increased EI by 25.9% (95% CI: 14.5, 37.3%; *p* = 0.001) and EBF by 14.6% (95% CI: 3.77, 25.5%; *p* = 0.010). Similarly, the intervention gave higher breastfeeding attitude scores (Effect size (ES): 0.85SD; 95% CI: 0.70, 0.99SD; *p* < 0.001), but not higher knowledge scores (ES: 0.15SD; 95% CI: −0.10, 0.41SD; *p* = 0.173). From the several growth and morbidity outcomes evaluated, the only outcomes with significant intervention effect were a higher mid-upper arm circumference (ES: 0.25cm; 95% CI: 0.01, 0.49cm; *p* = 0.041) and a lower prevalence of respiratory infection (ES: −6.90%; 95% CI: −13.3, −0.61%; *p* = 0.033). Training WDA leaders to provide BFESI substantially improves EI and EBF practices and attitude towards breastfeeding.

## 1. Introduction

Globally, 2.5 million neonatal deaths occurred in 2017, accounting for 46% of all under-five mortality [1]. The majority of the global burden of neonatal mortality occurred in low- and middle-income countries (LMICs), and mainly from preventable causes. If scaled up at a universal level, breastfeeding can improve the survival of children by preventing an estimated 823,000 annual deaths in under-five children, of which 87% occurs in infants under 6 months of age [2]. Despite the well-established benefits of breastfeeding, only 42% of newborns globally initiate breastfeeding within 1 h [3], and only 37% of infants younger than 6 months in LMICs are exclusively breastfed [2].

A recent review concluded that the scale-up of breastfeeding protection, promotion, and support interventions is among the key strategies to achieve nutrition targets under the Sustainable Development Goals [2]. In this regard, community-based intervention approaches, including household service delivery, have been identified as particularly effective for scaling-up breastfeeding promotion and support interventions and reaching the populations at most risk [4]. Moreover, a systematic review of breastfeeding promotion interventions in LMICs found that peer-led support strategies were effective in improving the rates of exclusive breastfeeding (EBF), while the provision of education alone was not effective [5]. It has also been shown that community-based peer-support is an effective approach in increasing the rates of early initiation of breastfeeding (EI) and the duration of EBF in populations in LMICs [6]. In Sub-Saharan Africa, a few studies that evaluated the effectiveness of community-based peer-support interventions reported increased optimal breastfeeding practices [7,8,9].

The World Health Organization recommends EI of breastfeeding within 1 h of birth, EBF during the first 6 months of life, and continued breastfeeding at least until the age of 2 years [10]. In Ethiopia, only 58% of infants were exclusively breastfed for six months in 2016, giving a national average duration of EBF of 3.1 months [11]. Furthermore, there has been limited progress in improving optimal breastfeeding practices in the country. For instance, the National Nutrition Program set a target to increase the rate of EBF by 22% between 2016 and 2020—only a 1% increase was achieved by 2019 [12,13]. Therefore, there is a critical need to identify effective strategies for large-scale implementation of breastfeeding promotion intervention in the country.

The Ethiopian perinatal care packages provided at health facilities include four focused antenatal visits, delivery care, and five contacts of postnatal care [14]. As part of newborn care immediately after birth, health workers encourage women to initiate breastfeeding within one hour and counsel them on correct positioning. After discharge from a health facility, health workers advise women to exclusively breastfeed their baby for six months during each postnatal contact. The challenge for implementing health facility-based breastfeeding promotion intervention is that most rural Ethiopian women deliver at home with very few women making postnatal visits. For instance, the 2016 EDHS reported that institutional delivery was only 20% among women living in rural areas [11]. On the other hand, the current community-based nutrition program in Ethiopia includes an outreach strategy to reach populations living in rural areas through health extension workers (HEWs) and local women peer-educators—also known as the Women Development Army (WDA) leaders [13]. The main components of the HEW program provided as routine care include a monthly growth monitoring and promotion for children under 24 months, biannual vitamin A supplementation and deworming, and a quarterly screening for acute malnutrition. Furthermore, HEWs are expected to provide infant and young child feeding counselling to mothers during the monthly growth monitoring sessions and during antenatal and postnatal care at health posts or through home visits though they do not have a fixed schedule for this. However, the HEWs are overburdened with their assigned tasks [15], which hinder them from providing the support lactating mothers need, particularly during the immediate postpartum period when women may give up breastfeeding due to difficulties they may face.

In Ethiopia, large-scale intervention studies such as the Alive and Thrive project [16] used an evaluation design with no control groups, making it unable to draw causal inference; and the “CBMNH-N multi-country project logic models focus on pregnant women and their newborns” employed a quasi-experimental design lacking true randomization [17]. The Muskoka Initiative Consortium—Knowledge Management Initiative focused on identifying contextual factors associated with the successful implementation of EBF programs [18] while in a trial in Hawassa, the intervention involved a single prenatal educational session [19]. None of these prior interventions combined prenatal and postnatal support with high intensity (>5 visits) which is found to have the highest impact on EBF in developing countries. Moreover, none of these interventions involved extra visits for women who experienced breastfeeding problems when they most needed the support to continue EBF. We, therefore, designed a cluster-randomized controlled trial evaluating the effectiveness of a community-based peer-led breastfeeding education and support intervention delivered during the prenatal and postnatal period through the established WDA system.

The primary outcomes of the trial were early initiation, exclusive breastfeeding, and infant growth, while childhood morbidity, breastfeeding knowledge, and attitude were secondary outcomes. The reason why we selected breastfeeding practices (early initiation and exclusive breastfeeding) as a primary outcome is because these two practices are associated with enormous health benefits, such as the reduced risk of morbidity and mortality. Improved breastfeeding practice is also associated with improved growth of infants; however, there are conflicting findings among the available evidence and thus growth was also selected as a primary outcome. As a secondary outcome, we wanted to see if the improved breastfeeding practice is associated with a reduced prevalence of childhood illnesses. Moreover, breastfeeding knowledge and attitude are an intermediate effect of the intervention, which could contribute to the improved practice and thus we identified them as secondary outcomes.

## 2. Materials and Methods

### 2.1. Study Design and Setting

The design and methods used in this trial, the Breastfeeding Education and Support Intervention (BFESI), are described in detail elsewhere [20]. Briefly, the study involved a cluster randomized, parallel-group, single-blinded trial evaluating the efficacy of BFESI on EI and EBF practices, and infant growth in a rural Ethiopian setting. The study was conducted in the Manna district located in Jimma Zone in southwest Ethiopia, where there was no similar ongoing intervention or project. From the total of 78 sub-districts under Mana, 36 sub-districts were selected for the study. The 36 sub-districts selected for the study were randomly assigned to either an intervention group (*n* = 18) receiving the BFESI or a control group (*n* = 18) receiving the routine Ethiopian healthcare service. We used simple randomization with a 1:1 allocation to allocate sub-districts to either control or intervention. First, the 36 sub-districts were listed alphabetically and then they were sequentially numbered starting from 01 to 36. Then we generated 18 random numbers from those 01 to 36 using MS Excel 2010 and the districts with the selected random numbers were assigned to the intervention group, while the rest were assigned to the control group. The generation of the allocation sequence and the randomization of clusters were done by a statistician blinded to study groups and not participating in the research. Allocation concealment was not done for study participants, as they would know if they were in the intervention group or not. However, data collectors were masked to the sub-district allocation by not being informed of the allocation, not being part of trial implementers, and not being residents in any of the sub-districts.

### 2.2. Participants

All pregnant women in the selected sub-districts were identified by reviewing the HEWs’ antenatal care logbook. Women in their second or third trimester of pregnancy, who were willing to participate with no intention of leaving the study area during the intervention period, were enrolled for the study between May and September 2017. Study exclusion criteria were the presence of severe mental illness that could interfere with consent and study participation, serious illness or clinical complications warranting hospitalization, the occurrence of maternal death, abortion, stillbirth, infant death, twin gestation, preterm birth (at <37 weeks gestation), or any child congenital malformation that could interfere with breastfeeding.

### 2.3. Procedures

#### 2.3.1. Training of Peer Supporters

The Ethiopian government introduced the health extension program in 2003 aimed at improving access to primary health care to rural communities through the expansion of health posts and training of HEWs [21]. After training and deployment to health posts, HEWs train model families on 16 health extension program elements over several weeks for 96 h. A woman who knows all the 16 packages and practices them is selected from the model family to lead other five women in her neighbourhood, supporting their adaptation of good practices, such as vaccinating their children, sleeping under mosquito bed-nets, building separate latrines, and using family planning [22]. Peer support is defined as the provision of emotional, appraisal, and informational assistance by a created social network member who possesses experiential knowledge of a specific behaviour or stressor, similar characteristics as the target population, and the ability to address a health-related issue [23]. With this in mind, for this trial, we selected WDA leaders who could read and write the local language, aged 24–39 years, with experience of motherhood as well as breastfeeding, being from the same community as the women they should support.

The WDA leaders from the intervention communities were trained for five days as breastfeeding peer-supporters by a nutritionist and a nurse with prior training on breastfeeding. WHO/UNICEF/USAID manuals were used to develop Ethiopian training guidebooks in the Afan Oromo language [24,25,26]. Moreover, to equip WDA leaders with the ability to educate and support study participants, a handbook with counselling cards were translated and prepared from these manuals. The training involved classroom lectures, demonstrations, and role play. Use of the manual and the counselling cards was practised through role-playing in teams with feedback from peers. Follow-up and supervision were carried out monthly during scheduled visits, in addition to unannounced spot-checks. Every pregnant mother was given a form to tally the number and timing of the visits she received from the WDA leaders. During the supervision, the supervisors checked the tallied paper and collected it at the end of the intervention.

#### 2.3.2. Breastfeeding Education and Peer Support

Peer-supporters made home visits to women in the intervention clusters according to a pre-specified schedule [20]. During pregnancy, they made two home visits in the last trimester of pregnancy: during the 8th and 9th month. Visits after delivery were scheduled on the 1st or 2nd, 6th or 7th and 15th day, and thereafter monthly until the infant was five months. During the two antenatal visits, peer-supporters encouraged delivery at the nearby health centre, emphasized the importance of initiating breastfeeding within 1 h of delivery, feeding colostrum first, discouraging the use of traditional pre-lacteal foods (items given to newborns before breastfeeding is established such as raw butter, plain water and milk-other than breast milk), and post-lacteal foods in addition to advising them to eat one extra meal during pregnancy to support lactation. The discussions were combined with the use of educational materials and practical demonstrations on proper breastfeeding positioning and latching. During the first two weeks after delivery, peer-supporters emphasised frequent and on-demand breastfeeding, encouraged stopping any traditional pre-lacteal foods or post-lacteal food items if already given to the child. Besides, peer-supporters observed the positioning, latching, and feeding of the newborn, solving any breastfeeding problems and providing appropriate feedback, while encouraging the mothers to continue EBF for six months. During these visits, women were advised to eat two extra meals during lactation from a variety of foods available in their area to provide energy and nutrition for themselves and their babies as well as to secure sufficient breast-milk production. Starting from month one, in addition to the above components, peer-supporters emphasized techniques for preparing for work and management of breast-milk (breast-milk expression, storing breast-milk), discussed the lactation amenorrhea method, and other family planning options. Hands-on guidance was provided only when necessary. Personal cleanliness and domestic hygiene, hand washing before feeding, after going to the toilet, and after changing babies’ diapers, were promoted during each visit. The mothers were encouraged to ask questions related to any topic discussed. Peer-supporters also provided additional visits if women experienced breastfeeding problems such as engorgement, cracked nipple or insufficient breastmilk that prohibited them from continuing breastfeeding. In addition to the informational support described above, women also received an emotional, appraisal, and instrumental support (Appendix A). The duration of each visit was typically 20–40 min.

Women in the control group received the routine care offered by the HEWs and WDA leaders working in their cluster, similar to that received by women in the intervention group [22]. The current Ethiopian standard/routine prenatal and postnatal care by HEWs includes providing four focused prenatal visits, developing an individualized birth preparedness and complication readiness plan, accompanying a woman to a health facility during delivery, and conducting four postnatal visits [27]. Moreover, as part of the community-based nutrition program, HEWs are expected to deliver the following key breastfeeding and nutrition messages to mothers during the monthly growth monitoring sessions or during antenatal or postnatal care visits: the importance of antenatal care, maternal nutrition during pregnancy and breastfeeding, early initiation of breastfeeding, proper positioning and attachment, EBF for six months, breastfeeding on demand, and complementary feeding [24]. WDA leaders also support the HEWs by educating and mobilizing communities to use key available health services, including dissemination of essential health messages such as infant and young child feeding practices.

### 2.4. Outcome Measures

The primary study outcomes were rates of EI and EBF for six months and infant growth. Secondary outcomes included maternal knowledge and attitude towards breastfeeding at the endline. We further included morbidity for common childhood illnesses over the past two weeks as an additional secondary outcome, although this was not considered a priori in the study protocol.

### 2.5. Data Collection

Data were collected at three time-points including at study enrolment (May-September 2017), at around 1 month (±2 weeks), and 6 months (±2 weeks) postpartum. At baseline, data on demographic and socio-economic characteristics, information on various maternal factors, and maternal knowledge and attitude towards breastfeeding were assessed. At one month postpartum, information about pregnancy outcome and other study exclusion criteria, and maternal practice on early initiation of breastfeeding including information about colostrum and pre-lacteals feeding were gathered. Data collected at around six months postpartum included maternal knowledge and attitude towards breastfeeding, EBF practice, infant anthropometry measurements, and morbidity. Data were collected by trained nurses and all instruments used were Afan Oromo language translations of English versions.

Gestational age was determined based on the last menstrual period (LMP). LMP was self-reported at baseline during enrolment. If women did not remember the exact date of the month, the 15th day of the month was used. First, we determined the estimated delivery date from the LMP and then subtracted the difference between the estimated delivery date and the actual delivery date from 280 days. Finally, we divided the total number of days by 7 to determine the gestational age in weeks. Maternal knowledge and attitude towards breastfeeding were assessed using Afan Oromo (AO) versions of the Breastfeeding Knowledge Questionnaire (BFKQ) and the Iowa Infant Feeding Attitude Access Scale (IIFAS), which were culturally adapted and validated in the same population. Details of the adaptation process and psychometric properties of both tools are reported previously [28]. Both the BFKQ-AO and the IIFAS-AO had an acceptable level of internal consistency with Cronbach alpha values of 0.79 and 0.72, respectively. Since our breastfeeding knowledge questionnaire adopted from Malaysia does not have a cut-off point suggesting an optimal knowledge level, we used a cut-off of ≥the median for a good level of knowledge. For attitude, we used the recommended cut-off of ≥70 scores for a positive attitude towards breastfeeding [29]. EI of breastfeeding and EBF practices were defined according to the WHO Infant and Young Child Feeding indicators [30]. Accordingly, mothers were asked how soon after delivery they put their newborn to the breast with responses ≤1 h considered as optimal EI practice. EBF practice was defined as feeding infant no other food or drink, not even water, except breast milk for the first six months of life, but allowing the infant to receive oral rehydration solution, drops, and syrups (vitamins, minerals, and medicines). The following questions were asked to evaluate EBF practice: (i) For how many months did you exclusively breastfeed (name); (ii) Are you currently giving your infant any food/drink other than breast milk? (iii) If yes to ii, we asked the age at which the food/drink was started. Thus, we used questions i, ii, and iii to determine if the child had been exclusively breastfed for six months since birth.

Anthropometry measurements of infant length, weight, and mid-upper arm circumference (MUAC) were done in duplicate by two independent teams of data collectors and recorded on separate forms so that the first measurement could not influence the second. Then, a supervisor compared the duplicate measurements, and both teams repeated the measurement whenever there was a difference of ≥0.5 kg for weight, ≥1.0 cm for length, or ≥0.5 cm for MUAC. Recumbent length was measured using a length board (SECA 417) with a precision of 0.1 cm. Weight was measured together with the mother using an electronic scale (SECA 876) to the nearest 1.0 g. MUAC measurement was taken on the left arm to the nearest 0.1 cm using flexible non-stretchable measuring tapes (SECA 212). Instruments were calibrated before each measurement session. The average value of the duplicate anthropometry measurements was used for analysis, and length-for-age (LAZ), weight-for-length (WLZ), and weight-for-age (WAZ) z scores were calculated based on the WHO 2006 Child Growth Standards using the Stata zscore06 command [31]. Child stunting, wasting, and underweight were determined from the respective z-score values using a cut-off <−2 SD from the median.

Mothers were asked to recall infant morbidity during the two weeks before the endline follow-up. Diarrhoea was defined as three or more liquid or semisolid stools within a day. Fever was determined by the mother’s report. Acute respiratory infection was defined as a combination of fever and cough. Serious illness was the occurrence of an illness that required medical attention, i.e., hospital or health centre visits. Household wealth status was assessed using a 16-item household asset questionnaire adapted from the Ethiopian Demographic and Health Survey [11], and principal components analysis was used to generate a household asset score. Household food security status was assessed using the Household Food Insecurity Access Scale from the Food and Agriculture Organization [32].

### 2.6. Statistical Analysis

The sample size was calculated by taking into account an assumed intracluster correlation coefficient, the expected effect, and the power of the study [20]. In Ethiopia, the current overall rate estimate of EBF was 58%, according to the EDHS 2016 report [11]. We hypothesized that the BFESI would increase the prevalence of EBF to 78% in the intervention group. To detect a 20% difference in the rates of EBF, with 80% power, and 5% type I error, assuming an intra-cluster correlation of ρ = 0.1 [9], 10 pregnant women were needed per study cluster. We inflated the sample size from 346 to 432 to accommodate for a potential 20% attrition rate.

Data were entered using Epi-data version 3.1 (EpiData Association) and consistency checks and statistical analysis were conducted using Stata version 13.0 (StataCorp LLC, College Station, TX, USA). Descriptive statistics were summarized as frequency and percentage for the categorical variables, and mean and standard deviation for the continuous variables. Maternal knowledge and attitude scores were standardized to z scores based on the distribution of the data. The effects of the intervention were estimated using linear regression models for the continuous outcomes, and linear probability models for the binary outcomes. The use of linear probability models for binary outcomes is well established and allows for a straightforward interpretation of the average intervention effect expressed as a risk difference using percentage points [33]. However, for the rare outcomes of child stunting, wasting, and underweight, we fitted logistic regression models. In all models, we applied a robust standard error estimation taking into account the clustering of subjects by sub-districts. Both unadjusted and adjusted group differences were estimated with covariates used for adjustment, including maternal age, educational status, wealth index, parity, household food insecurity status and IIFAS score at baseline. Analyses were done following the intention-to-treat principle. For this purpose, we conducted multiple imputations of missing data using chained equations under the assumption of missing at random. To estimate the regression coefficients, we ran a hundred imputations of missing data for cases lost-to-follow-up. Statistical significance was declared at *p*-value < 0.05.

## 3. Results

A total of 468 pregnant women (n intervention = 249; control = 219) were enrolled at baseline. At the one-month postpartum follow-up, 47 subjects (n intervention = 28; control = 19) were excluded because of newborn death (*n* = 11), twin deliveries (*n* = 4), stillbirth (*n* = 6), maternal death (*n* = 1), abortion (*n* = 3), or change in residence (*n* = 22) (Figure 1). At the six-month postpartum follow up, 12 subjects (n intervention = 9; control = 3) were excluded because of child death (*n* = 2) or change in residence (*n* = 10). Ultimately, we have outcome data from 421 (90.0%) mother-child pairs (n intervention = 221; control = 200) at the one-month postpartum follow-up and from 409 (87.4%) mother-child pairs (n intervention = 212; control = 197) at the six-month postpartum follow-up. Reasons for drop out are reported according to the CONSORT reporting guideline in Figure 1. Baseline characteristics of study participants are presented in Table 1.

Peer supporters’ visit coverage was reported by women and peer counsellors during the two follow-up visits. Among 221 women who were available for the first follow-up at month one after delivery, 189 (75.9%) received the two prenatal visits and 152 (80%) of the visits were timely; 32 (12.9%) women received only one prenatal visit as per the schedule. Out of 212 women who were available for the last follow-up visit, 150 (70.8%) received all the planned 8 visits and 119 (79.3%) of these visits were timely. However, 35 women received seven visits and 29 (82.9%) visits were timely, while 27 women received 6 visits with 20 (74.1%) of the visits being timely.

### 3.1. Effects on Breastfeeding Practices

A significantly higher proportion of newborns in the intervention group, 181 (72.7%), initiated breastfeeding within the first hour after delivery compared to the control group, 89 (40.6%) (*p* = 0.001) (Table 2). Similarly, EBF was significantly higher in the intervention group, 170 (68.3%), than in the control group, 120 (54.8%) (*p* = 0.009). In the adjusted analysis, BFESI significantly increased the rate of EI of breastfeeding by 25.4% (95% CI: 14.5, 37.3%; *p* = 0.001) and EBF by 14.6% (95% CI: 3.77, 25.5%; *p* = 0.010), as compared to the control group.

### 3.2. Effects on Infant Growth and Nutritional Status at the Age of Six Months

We found no statistically significant difference between the intervention and control groups on LAZ and WAZ or the prevalence of stunting, wasting, or underweight (Table 3). However, infants in the intervention group had a significantly higher MUAC than infants in the control group (effect size: 0.25 cm; 95% CI: 0.01, 0.49 cm; *p* = 0.041)).

### 3.3. Effects on Maternal Knowledge and Attitude towards Breastfeeding

BFESI resulted in a significantly higher maternal breastfeeding attitude score (effect size: 0.85 SD; 95% CI: 0.70, 0.99 SD; *p* < 0.001) and a higher proportion of mothers with positive attitude towards breastfeeding (30.4% (23.4, 37.4%); *p* < 0.001), as compared to the control group (Table 2). There was an increase in maternal knowledge level in both groups from baseline. The intervention resulted in a non-significant trend towards a higher maternal knowledge score (Effect size: 0.15 SD; 95% CI: −0.10, 0.41 SD; *p* = 0.173).

### 3.4. Effects on Infant Morbidity at Six Months of Age

During the two weeks before the six-month postpartum follow-up, we found no statistically significant group differences in the occurrence of common childhood illnesses, including serious illnesses, except that infants in the intervention group had a lower prevalence of cough with fever (Effect size (95% CI): −6.90% (−13.3, −0.61%); *p* = 0.033) (Table 3). In the complete-case analysis with only available information, similar results were obtained for the intervention effect in all outcomes (Appendix A).

## 4. Discussion

This study demonstrated that engaging WDA leaders, who already function in the healthcare system as peer-educators, successfully improved maternal practices related to EI of breastfeeding and EBF in rural Southwest Ethiopia. Peer-led education and support intervention significantly increased EI of breastfeeding by 26%, and EBF by 15%, compared to routine healthcare service. BFESI also resulted in a substantial improvement in maternal attitude towards breastfeeding. However, we did not find important impacts of the intervention on maternal knowledge about optimal breastfeeding practices, child growth or nutritional status, or child morbidity outcomes.

This study confirms that BFESI can ensure EI of breastfeeding within 1 h of birth and EBF. This is consistent with what has been reported by other studies [6,7,8,9,34]. The few RCTs from Sub-Saharan African countries, Kenya [8], rural Malawi [7], and the PROMISE EBF trial [9] concur with the presented study from Ethiopia.

The success of the peer-led interventions in improving EI, EBF, and attitude could be explained by several factors. Firstly, evidence from systematic review demonstrated that education alone is not effective and individual level combined prenatal and postnatal support with high intensity (>5 contacts) had the highest impact on EBF in developing countries [5,35]. In this trial, women received peer-support both during the antenatal and postnatal period with a minimum of 10 visits, with the majority of the women receiving all prenatal and postnatal visits as per the schedule. Moreover, the first two postnatal visits took place in the first week since this a critical time when women may give up breastfeeding due to difficulties they may face. Furthermore, the peers delivered instrumental/practical support during the scheduled/extra visits if the women experienced any breastfeeding problem. Thus, the timing and intensity of support received may have helped to build the mothers’ confidence, improve feeding technique, and prevent or resolve breastfeeding problems. Secondly, the WDA program is based on a theory of behaviour change where others in the community admire and copy the behaviour of these “model” women [36]. Furthermore, according to the theory of planned behaviour, people tend to perform certain behaviours when they believe that the “important others” think they should perform them [37]. Thus, owing to the fact that WDA leaders are from model households that are admired in the community, their function as a peer might have made them effective communicators and led to the success of the intervention.

Our findings on infant growth were similar to a previous study from India reporting the lack of effects of a breastfeeding promotion intervention on infant linear or ponderal growth [38]. A systematic review of breastfeeding promotion intervention on child growth suggests that it can lead to a modest reduction in body mass index or WAZ [39]. In our study, except for a modest increase in MUAC, the differences for LAZ, WAZ, and WLZ were not statistically significant. These findings could be due to the effect of exclusive or predominant breastfeeding that promotes higher accretion of fat mass during the first six months of life [40]. Moreover, not only breastfeeding behaviour but also intrauterine growth retardation contributes considerably to infant growth faltering, and growth faltering is also more prominent after six months of age when infants start complementary feeding [41]. On the other hand, it should be noted that our study was not adequately powered to detect potentially important differences in growth outcomes.

There is strong and convincing evidence that breastfeeding reduces the incidence and severity of acute infections, especially diarrhoea and lower respiratory tract infections such as pneumonia [2]. In a review by Chapman et al. [42], peer counselling was found to reduce the prevalence of diarrhoea. A trial from India revealed a similar effect [38]. In the current study, although the intervention group had a reduced prevalence of respiratory infection, we did not observe a lower prevalence of diarrhoea in our intervention group. This is in line with the PROMISE EBF trial and another study in Guinea-Bissau where diarrhoea was not significantly different between the intervention and control groups [9,43]. Although exclusive breastfeeding is one prevention strategy for diarrhoeal morbidity, it is not the only strategy [44]. Basic water, sanitation, and hygiene are among other strategies for diarrhoea prevention that are influenced by maternal age, educational status, occupation, and household living conditions [45]. However, these variables are similarly distributed between the intervention and control clusters, which might explain the absence of statistical difference in diarrhoeal prevalence between the two groups. Moreover, the diarrheal disease has an association with seasons, where some pathogens are more prevalent during rainy seasons contaminating water sources [46]. In this regard, although most endline data were collected during rainy seasons, diarrheal illness was statistically different between the groups when adjusted for the season.

Our intervention aimed to improve maternal knowledge and attitudes towards optimal breastfeeding practices. The level of breastfeeding knowledge increased in both the intervention and control group, although there was no statistical difference between the two groups. However, the intervention significantly improved the attitudes towards breastfeeding. For a behaviour change, not only knowledge and personal intention to act, but also the availability of social support are important [47]. Moreover, as reflected in the theory of planned behaviour, the peer-led education and support provided by the WDA leaders might have influenced the women to develop a positive attitude and a greater sense of confidence and commitment, leading to the adoption of the behaviour even without much gain in knowledge [37,48].

Our study has several strengths. The study had a high response rate with few dropouts. We used internationally developed breastfeeding knowledge and attitude tools that were validated in the Ethiopian setting. Besides, for the majority of the women, all prenatal and postnatal visits were implemented as per the schedule. Further, unlike peer-support interventions conducted in sub-Saharan Africa that selected peer counsellors who were not part of the healthcare system, training WDA leaders who are already introduced into the healthcare system by the government to support HEWs, as peer-supporters, may provide the opportunity to integrate this intervention into the existing healthcare system.

Our study had some limitations related to the outcome measurements. First, the allocation status of the clusters was concealed only from the outcome assessors since it was not possible to blind the participants due to the nature of the study. This might have introduced social desirability bias in outcomes such as breastfeeding practice and morbidity outcomes that were assessed through maternal self-report. Measuring social desirability to assess for potential biases in our estimates would have confirmed the accuracy of self-reported outcome measures. Second, reporting breastfeeding indicators based on both point-in-time and life-long data has been recommended as a stronger approach than either of these in isolation [49]. In the present trial, although we measured breastfeeding practices using a 24-h recall, a week recall, and a since birth, we used the since birth data to calculate EBF. We believe that due to within-subject variance, assessing practice over the past 24 hours is not superior to using a since birth measure. Third, additional assessment of EBF at month one follow-up would have enabled us to have a better picture of how EBF declined over time in our study population, and how this was affected by the intervention. Fourth, our sample size was limited in detecting small effects on child growth outcomes. Fifth, although there may be a possible risk of recall bias, any threats to validity related to the precise estimation of the outcomes affected both groups equally since we used a randomized design. Lastly, although assessing uptake of the intervention through qualitative study both from the perspectives of the mothers as well as peer-educators could have informed a future scale-up, this was not done. Future studies accompanying the scale-up should evaluate the quality, uptake, and cost-effectiveness of the peer-support interventions to map the sustainability of the intervention. Additionally, we have noted a high prevalence of food insecurity in the households of both the intervention and control groups. Therefore, we recommend that nutrition-sensitive interventions that can improve food security be implemented with future community-based breastfeeding promotion interventions.

## 5. Conclusions

In conclusion, the present study confirmed that in Ethiopia, WDA leaders in the primary healthcare structure can be trained as peer-supporters to provide successful breastfeeding education and support during home visits to the women in their neighborhoods. Our findings call for future studies assessing the feasibility of integrating the current intervention into the existing Ethiopian primary healthcare system with potential desirable impacts.

## Figures and Tables

**Figure 1 nutrients-13-01204-f001:**
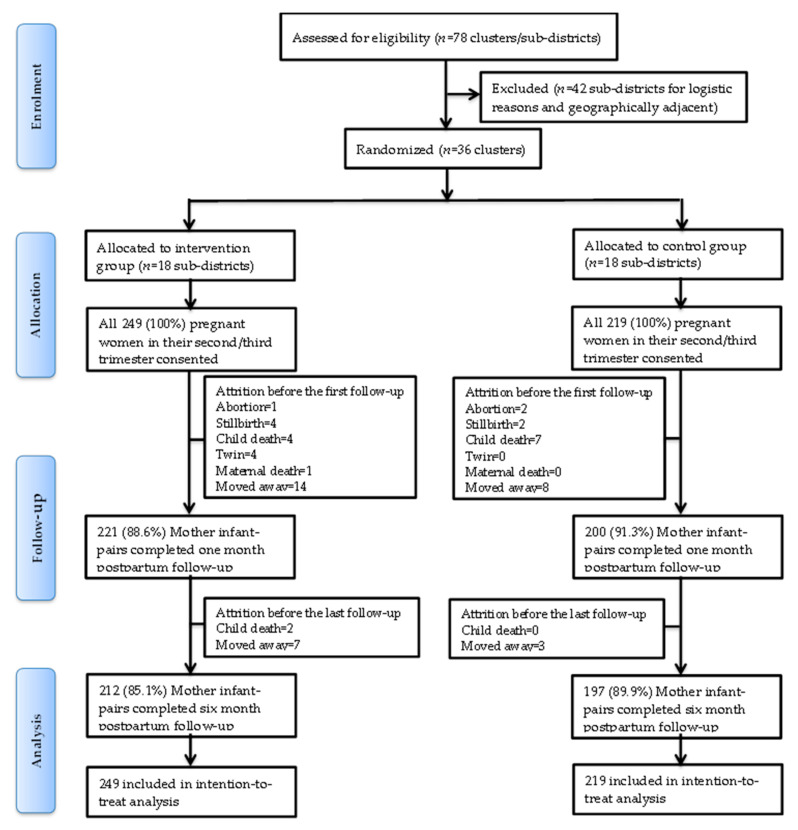
CONSORT (Consolidated Standards of Reporting Trials) flow diagram of the study.

**Table 1 nutrients-13-01204-t001:** Baseline characteristics of the study participants ^1^.

Variables	Intervention (*n* = 249)	Control (*n* = 219)
Maternal age, years		
15–19	8.84	12.8
20–34	85.1	81.3
35–40	6.02	5.94
Maternal educational status		
Illiterate	60.2	62.1
Can read and write	12.5	14.6
Primary education	7.63	4.57
Secondary education	19.7	18.7
Married	53.2	46.8
Housewife/farmer	93.6	94.1
Wealth quantiles		
Lowest	19.6	20.6
Second	18.1	22.4
Middle	18.9	21.0
Fourth	20.9	19.1
Highest	22.5	16.9
Birth interval ≥ 24 months	94.5	94.8
Primipara	18.5	18.3
Household food security status		
Food secure	57.0	61.2
Mildly food insecure	23.3	13.2
Moderately food insecure	12.5	17.4
Severely food insecure	7.23	8.22
Have a history of breastfeeding	94.3	93.5
IIFAS score	66.5 ± 6.93	64.8 ± 8.31
BFKQ score	24.7 ± 3.80	24.4 ± 4.81
Received ANC visit for the index baby	98.6	98.5
Received at least four ANC visits	50.6	53.3

^1^ Values are means ± SDs or proportions, ANC, Antenatal Care; BFKQ, Breast Feeding Knowledge Questionnaire; IIFAS, Iowa Infant Feeding Attitude Scale.

**Table 2 nutrients-13-01204-t002:** Breastfeeding knowledge, attitude, and practices by study arms ^1^.

Outcomes	Intervention (*n* = 249)	Control (*n* = 219)	Unadj diff (95% CI) ^2^	*p* ^2^	Adj Diff (95% CI) ^2^	*p* ^2^
Early initiation of breastfeeding ^3^	72.7	40.6	25.9 (13.8, 37.9)	<0.001	25.4 (14.5, 37.3)	<0.001
Exclusive breastfeeding ^4^	68.3	54.8	14.4 (3.91, 24.8)	0.009	14.6 (3.77, 25.5)	0.010
BFKQ score ^4^	26.3 ± 1.77	26.0 ± 2.07	0.15 (−0.09, 0.39)	0.211	0.15 (−0.10, 0.41)	0.173
IIFAS score ^4^	77.6 ± 9.04	67.7 ± 9.72	0.85 (0.70, 0.99)	<0.001	0.85 (0.70, 0.99)	<0.001
Good knowledge about breastfeeding ^4^	69.1	65.1	5.20 (−3.80, 14.2)	0.257	5.38 (−3.59, 14.4)	0.239
Positive attitude towards breastfeeding ^4^	75.5	43.8	30.4 (23.4, 37.4)	<0.001	30.4 (23.4, 37.4)	<0.001

^1^ Values are means ± SDs or proportions, ^2^ Unadjusted and adjusted group differences (CIs) and Ps estimated using linear regression models (as mean difference) for the continuous outcomes and linear probability models (as risk difference in percentage points) for proportions, with a robust variance estimation, used to account clustering of subjects by sub-districts. Covariates used for adjusted estimates were maternal age, educational status, wealth index, parity, and IIFAS at baseline. ^3^ Assessed at one month postpartum, ^4^ Assessed at six months postpartum, BFKQ, Breast Feeding Knowledge Questionnaire; IIFAS, Iowa Infant Feeding Attitude Scale.

**Table 3 nutrients-13-01204-t003:** Infant anthropometry and morbidity outcomes by study arms at 6 months postpartum follow-up ^1^.

Outcomes	Intervention (*n* = 249)	Control (*n* = 219)	Unadj Diff (95% CI) ^2^	*p* ^2^	Adj Diff (95% CI) ^2^	*p* ^2^
LAZ	−0.14 ± 1.15	−0.18 ± 1.22	0.05 −0.30, 0.39)	0.795	0.05 (−0.30, 0.39)	0.790
WAZ	0.01 ± 0.95	−0.12 ± 1.06	0.15 (−0.12, 0.41)	0.267	0.15 (−0.11, 0.41)	0.255
WLZ	0.23 ± 1.12	0.09 ± 1.07	0.15 (−0.16, 0.46)	0.326	0.15 (−0.15, 0.46)	0.311
MUAC	13.7 ± 0.96	13.5 ± 0.93	0.25 (0.00, 0.50)	0.048	0.25 (0.01, 0.49)	0.041
Stunted	4.82	6.39	−0.24 (−1.13, 0.64)	0.591	−0.25 (−1.11, 0.61)	0.565
Underweight	3.21	4.57	−0.31 (−1.29, 0.68)	0.537	−0.28 (−1.38, 0.82)	0.618
Wasted	4.02	1.37	1.16 (0.03, 2.29)	0.044	1.16 (0.00, 2.33)	0.051
Cough	16.9	24.2	−6.93 (−15.1, 1.29)	0.096	−7.09 (−15.1, 0.92)	0.081
Fever	16.1	21.5	−4.95 (−12.6, 2.69)	0.196	−4.71 (−12.2, 2.75)	0.207
Diarrhea	10.0	8.22	2.50 (−4.08, 9.09)	0.445	2.54 (−4.09, 9.18)	0.442
Fever with cough	10.0	15.5	−6.70 (−13.4, −0.03)	0.049	−6.90 (−13.3, −0.61)	0.033
Any illness	28.1	34.7	−6.07 (−16.4, 4.21)	0.238	−6.12 (−16.1, 3.85)	0.221
Serious illness	18.9	27.4	−8.17 (−17.4, 1.04)	0.080	−7.79 (−16.9, 1.29)	0.090

^1^ Values are means ± SDs or proportions, ^2^ Unadjusted and adjusted group differences (CIs) and Ps estimated using linear regression models (as means difference) for continuous outcomes, logistic regression models (as odds ratio) for nutritional status outcomes, and linear probability models (as risk difference in percentage points) for morbidity outcomes, with a robust variance estimation, used to account clustering of subjects by sub-districts. Covariates used for adjusted estimates were maternal age, educational status, wealth index, parity, IIFAS, and household food insecurity status at baseline, LAZ, length-for-age z scores; MUAC, mid-upper-arm-circumference in cm; WAZ, weight-for-age z scores; WHZ, weight-for-length z scores.

## Data Availability

The data presented in this study are available on request from the corresponding author, due to privacy restriction.

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
