# Peer review of "Breastfeeding Education and Support to Improve Early Initiation and Exclusive Breastfeeding Practices and Infant Growth: A Cluster Randomized Controlled Trial from a Rural Ethiopian Setting"

_nutrients, 2021, doi:10.3390/nu13041204_

Round 1
Reviewer 1 Report
This study exploited community-based peer-support interventions to improve breastfeeding practice in Ethiopia. Although the result suggests that the intervention is a promising tool to improve maternal-child health in Africa, there are several concerns as follows.
#1 Randomization
First, the 36 sub-districts were listed alphabetically and then a list of random numbers was generated in MS Excel 2010 and the generated values were fixed by copying them as “values” next to the alphabetic list of the sub-districts. After arranging the generated random numbers in ascending order, the first 18 sub-districts were selected as intervention clusters and the last 18 as control clusters.
I am concerned about this simple randomization method. Suppose if all the districts listed alphabetically bear some bias, applying random numbers to the order of the list may create selection bias. I recommend the authors to consult external statisticians for the appropriate way of randomization.
#2 Group comparison
Table 1. Baseline characteristics of study participant
Authors should present p-value to test different characteristics between the intervention group and the control group. The intervention group seems to have older age, a higher educational background which may play favorably toward significant results. If there are significant differences observed between the comparison groups, some amendments should be performed for example, by performing multiple logistic regression with the outcome of interest while entering intervention/control and covariates in the same models.
#3 Abstract
>A higher mid-upper arm circumference (ES: 0.25 cm; 95% CI: 0.01, 0.49 cm;P=0.041) and a lower prevalence of respiratory infection (ES: -6.90%; 95% CI: -13.3, -0.61%; P=0.033) were significant intervention effects related to growth and morbidity outcomes.
⇒I am wondering if these sentences are necessarily placed here because there are more numbers of items investigated which turned out to be nonsignificant. For example, the majority of Table 3 (Infant anthropometry and morbidity outcomes by study arms at 6 months postpartum follow-up) are not statistically significant.
Hence, the abstract should be neutral and fair to explain what they found in this study.
Reviewer 2 Report
This manuscript is well-written. I have some minor comments to improve the manuscript.
Introduction
- Please add the rationale for primary versus secondary outcomes.
- Please describe the current perinatal care package for women in Ethiopia, including breastfeeding education and health care. Please link the gap in health care to your intervention components. What is the ordinary number of visits and contents?
Methods
- How did you select 36 out of the 78 sub-districts?
- Routine care (control) needs to be described in more details. For example, what is the related breastfeeding and nutritional support contents? For the intervention group, was the intervention a replacement for routine care or in addition to routine care?
- Please add internal consistency or other psychometric property indicators if available for your knowledge/attitude scales.
- How did you combine "intention to treat principle" with handling missing data?
Results
- For those with food insecurity problems and/or mothers with inadequate nutrition, did you provide any assistance, regardless of intervention status?
- Please add some quality control results if you could. For example, actual number of visits received and actual time of visits.
Discussion
- Please take the quality control issue (implementation) into consideration in your discussion.
- In the context of food insecurity, do you recommend other measures to complement your program?
